# Stress and Nasal Allergy: Corticotropin-Releasing Hormone Stimulates Mast Cell Degranulation and Proliferation in Human Nasal Mucosa

**DOI:** 10.3390/ijms22052773

**Published:** 2021-03-09

**Authors:** Mika Yamanaka-Takaichi, Yukari Mizukami, Koji Sugawara, Kishiko Sunami, Yuichi Teranishi, Yukimi Kira, Ralf Paus, Daisuke Tsuruta

**Affiliations:** 1Department of Dermatology, Osaka City University Graduate School of Medicine, Osaka 5458585, Japan; mikay0115@gmail.com (M.Y.-T.); cebu-yuka@hotmail.co.jp (Y.M.); dts211@gmail.com (D.T.); 2Department of Otolaryngology and Head and Neck Surgery, Osaka City University Graduate School of Medicine, Osaka 5458585, Japan; kishiko@med.osaka-cu.ac.jp (K.S.); yteranishi@med.osaka-cu.ac.jp (Y.T.); 3Department of Research Support Platform, Osaka City University Graduate School of Medicine, Osaka 5458585, Japan; yukimi@med.osaka-cu.ac.jp; 4Dr. Phillip Frost Department of Dermatology & Cutaneous Surgery, University of Miami Miller School of Medicine, Miami, FL 33136, USA; rxp803@med.miami.edu; 5Centre for Dermatology Research, University of Manchester, and NIHR Manchester Biomedical Research Centre, Manchester M13 9PR, UK; 6Monasterium Laboratory, 48149 Münster, Germany

**Keywords:** CRH, mast cell, nasal mucosa, psychological stress, stem cell factor (SCF)

## Abstract

Psychological stress exacerbates mast cell (MC)-dependent inflammation, including nasal allergy, but the underlying mechanisms are not thoroughly understood. Because the key stress-mediating neurohormone, corticotropin-releasing hormone (CRH), induces human skin MC degranulation, we hypothesized that CRH may be a key player in stress-aggravated nasal allergy. In the current study, we probed this hypothesis in human nasal mucosa MCs (hM-MCs) in situ using nasal polyp organ culture and tested whether CRH is required for murine M-MC activation by perceived stress in vivo. CRH stimulation significantly increased the number of hM-MCs, stimulated both their degranulation and proliferation ex vivo, and increased stem cell factor (SCF) expression in human nasal mucosa epithelium. CRH also sensitized hM-MCs to further CRH stimulation and promoted a pro-inflammatory hM-MC phenotype. The CRH-induced increase in hM-MCs was mitigated by co-administration of CRH receptor type 1 (CRH-R1)-specific antagonist antalarmin, CRH-R1 small interfering RNA (siRNA), or SCF-neutralizing antibody. In vivo, restraint stress significantly increased the number and degranulation of murine M-MCs compared with sham-stressed mice. This effect was mitigated by intranasal antalarmin. Our data suggest that CRH is a major activator of hM-MC in nasal mucosa, in part via promoting SCF production, and that CRH-R1 antagonists such as antalarmin are promising candidate therapeutics for nasal mucosa neuroinflammation induced by perceived stress.

## 1. Introduction

It has long been appreciated that psychological stress may trigger or aggravate allergic rhinitis and other allergic diseases [1,2,3,4,5]; however, the underlying mechanisms are poorly understood. Although allergic rhinitis impairs patients’ quality of life, and prenatal maternal psychological stress might even predispose the offspring to postnatal rhinorrhea and nasal inflammation [6,7], there is no specific treatment for stress-aggravated nasal allergy. Here, we examined the hypothesis that the key neuroendocrine mediator of perceived stress, corticotropin-releasing hormone (CRH) [8,9,10], and its recognized function as a mast cell (MC) secretagogue [11,12,13], may play an as-yet underappreciated, critical role in stress-aggravated nasal mucosa inflammation and is an important target for therapeutic intervention.

CRH is the most proximal, key regulator of the central hypothalamic–pituitary–adrenal (HPA) stress response axis, which ultimately controls the production and secretion of glucocorticoids in the adrenal gland [8]. CRH also exerts multiple other functions, such as inhibiting proliferation and inducing the differentiation of epidermal keratinocytes [14] stimulating proliferation of dermal fibroblasts and melanocytes [15], and production of proinflammatory cytokines [16,17]. Importantly, human peripheral tissues also express functional HPA equivalents, including CRH, adrenocorticotropic hormone, and cortisol, and provide negative feedback regulation [18,19,20,21]. This has been documented best in human skin, where CRH may exert a dual role: short-term pro-inflammatory actions (e.g., via the induction of mast cell (MC) degranulation) and longer lasting immunoinhibitory functions via the stimulation of local cortisol synthesis by keratinocytes [12,22]. In addition to being produced by resident skin cell populations, such as keratinocytes and MCs, CRH can be released from sensory skin nerves and MCs in response to psychoemotional and environmental stressors [17,19,23].

Much less is known about the function of CRH in human mucosal tissues that express CRH receptors (CRH-Rs, type 1 (CRH-R1 and type 2 (CRH-R2)) [24]. Limited information is available on the impact of CRH-R agonists on human intestinal and bladder mucosa [24,25,26,27,28,29], but the function of CRH-R-mediated signaling in human airway mucosa remains unknown. Reportedly, CRH-R protein expression is upregulated in allergic human nasal mucosa [30], although its functional relevance is unknown. Circumstantial evidence has encouraged the hypothesis that perceived stress may aggravate bronchial asthma via a stress-induced increase in serum CRH levels, which activate lung MCs [31]. This corresponds to the observation that both atopic dermatitis and psoriasis patients, whose skin lesions can be aggravated by psychoemotional stress [32,33], have increased serum CRH levels [13,34] and that CRH protein expression and CRH-R1 transcription are increased in psoriatic skin lesions [35,36]. Perceived stress increases MC degranulation in rodent skin [37] and stress-induced neurogenic skin inflammation in mice are largely mitigated in substance P-receptor- or MC-deficient mice [38]. Furthermore, acute stress can activate skin MCs via increased intracutaneous CRH expression [37,39,40], and intradermal administration of CRH activates rat connective tissue-type MCs (CT-MCs) via CRH-R1, leading to vasodilation and increased vascular permeability [41,42]. Finally, CRH-R1-mediated signaling enhances the MC degranulation response in mice [43], whereas restraint-stress-induced MC activation in rat skin is inhibited by CRH-blocking antibodies [37].

If the hypothesis that human airway mucosa MCs are activated under conditions of perceived stress via CRH/CRH-R stimulation is confirmed in situ, CRH and CRH-Rs would become promising novel targets for therapeutic interventions in a wide range of stress-aggravated, MC-dependent human airway diseases ranging from bronchial asthma via allergic rhinoconjunctivitis to nasal polyposis [44,45,46,47]. Confusingly, however, a murine asthma model showed that CRH deficiency enhances allergen-induced airway inflammation, leading the authors to speculate that inherited or acquired CRH deficiency could increase asthma severity in patients [48]. In addition, in mice, CRH-R2 stimulation actually inhibits perceived stress-induced MC degranulation and limits the severity of immunoglobulin E-mediated anaphylaxis [49]. Therefore, robust evidence is urgently needed to determine whether and how CRH impacts unmanipulated, primary, tryptase+ human nasal mucosa MCs (hM-MCs) within the physiological tissue.

To address this question, we were guided by two previously published human organ culture studies. First, ex vivo, CRH not only induces the degranulation of CRH-R+ human skin MCs in the connective tissue sheath of scalp hair follicles (HFs), but also promotes their intracutaneous differentiation from resident c-Kit+ progenitor cells. CRH also upregulates stem cell factor (SCF) production by the HF epithelium, whereas CRH-induced human skin MC degranulation is inhibited by SCF-neutralizing antibody (anti-SCF) [11]. Second, serum-free organ culture of human nasal polyp (NP) tissue provides an excellent surrogate assay for studying the response of primary hM-MCs to neuro-endocrine stimuli and for dissecting the role of SCF in these responses. Moreover, gene silencing is possible in NPs ex vivo, facilitating mechanistic research under human organ culture conditions [50]. Therefore, we employed this instructive and clinically relevant human upper airway mucosa assay to clarify whether CRH-R stimulation with the central neuroendocrine stress mediator CRH activates or negatively regulates hM-MCs in situ, and whether CRH-induced SCF production by the NP epithelium plays a role in the observed MC effects.

To examine the in vivo relevance of the generated human ex vivo data, we also performed a limited pilot assay investigating whether restraint stress [51] activates mice M-MCs (mM-MCs) and determined whether any effect is CRH-dependent by intranasally applying the selective CRH-R1 antagonist antalarmin [52].

Taken together, these analyses demonstrate that CRH stimulates the degranulation and proliferation of unmanipulated, primary hM-MCs within human nasal mucosa ex vivo in a CRH-R1- and SCF-dependent manner and suggest that stress-induced M-MC degranulation in vivo is also CRH-R1-dependent and can be inhibited with antalarmin.

## 2. Results

### 2.1. c-Kit or Tryptase+ Human Nasal Mucosa MCs Expressed CRH-R1s In Situ

We previously reported that human skin MCs express CRH-R1 in situ [11]. As MC phenotype and function are dictated by the local tissue milieu, including the neuroendocrine signals arising from it, and the multi-directional site-specific signaling network in which they are embedded [12,44,50,53], it is vital to analyze MC biology within resident tissues to preserve MC–epithelial interactions. NPs have long been used as a surrogate tissue for human upper airway mucosa [54], where primary hM-MCs can be studied within their physiological tissue in the absence of confounding neural and perfusion-dependent systemic inputs [50].

Therefore, we used this serum-free, clinically relevant NP organ culture assay to investigate whether tryptase+ or c-Kit+ hM-MCs in paraffin sections of human NPs co-express CRH-Rs using double immunofluorescence microscopy. Tryptase is one of the main mucosal-type MC proteases that is critically involved in allergic inflammation [55], whereas c-Kit can be used as a marker for detecting both immature and mature MCs [56,57]. This showed that both c-Kit+ and tryptase+ hM-MCs co-express specific immunoreactivity for CRH-R1 in situ (Figure 1a,b), suggesting that, similar to their counterparts in human skin [11,53], both less-differentiated c-Kit+ and mature tryptase+ hM-MCs are capable of responding to CRH stimulation. In contrast with intestinal mucosal MCs [58], tryptase+ hM-MCs hardly expressed CRH-R2 in situ (Figure 1c). Moreover, NP epithelium expressed CRH protein (Figure 1d) as well as CRH-R1 and exhibited hardly any CRH-R2 immunoreactivity (Figure 1e).

### 2.2. CRH Increased the Number and Degranulation of Tryptase+ hM-MCs In Situ

Because we previously showed that CRH both increases the number and induces the degranulation of human skin MCs ex vivo [11], we were curious to learn how hM-MCs would respond to the same dose and short-term incubation with CRH (10^−7^ M, 24 h) in human NP organ culture. Toluidine blue histochemistry revealed that, just as in human perifollicular skin MCs [11], CRH rapidly and significantly increased both the number and percentage of degranulation of hM-MCs in situ (Figure 2a). We also confirmed this finding by tryptase immunohistochemistry, which revealed that CRH significantly increased the number of tryptase+ hM-MCs (Figure 2b,c). Further evaluation of CRH-induced hM-MC degranulation by quantitative tryptase immunohistomorphometry (Figure 2d,e) and electron microscopy (Figure 2f) in situ confirmed that CRH increased the percentage of degranulated hM-MCs (Figure 2e). The overall increase in MC granule size reportedly reflects MC maturation [59,60], and MC granules enlarge immediately before degranulation [61]. CRH treatment significantly increased the diameter of tryptase+ granules within hM-MCs over 24 h of NP organ culture (Figure 2g).

However, in striking contrast to native human skin MCs ex vivo [11], CRH did not significantly alter the number of c-Kit+ or chymase+ hM-MCs in NPs (Appendix A). This suggested a selective effect of CRH on mature, tryptase+ hM-MCs, and that—other than in human skin [11]—CRH does not promote the differentiation of immature, c-Kit+ resident MC progenitors into tryptase+ hM-MCs. This left the CRH-induced proliferation of tryptase+ hM-MCs as the most reasonable explanation for the significantly increased number of these cells in CRH-stimulated NPs.

### 2.3. CRH Stimulated the Intramucosal Proliferation of Tryptase+ hM-MCs

To probe how CRH affects hM-MC proliferation and apoptosis in situ, we performed tryptase/Ki-67 double immunostaining. Quantitative immunohistomorphometry revealed a significant increase in the percentage of tryptase/Ki-67 double-positive mature hM-MCs in CRH-treated NPs (Figure 3a). This was independently confirmed by proliferating cell nuclear antigen (PCNA), another cell proliferation marker [62], via tryptase/PCNA double immunostaining (Figure 3b). As we previously showed that perifollicular MCs proliferate in human skin in situ under inflammatory conditions (alopecia areata) [63], this further questions the often-repeated dogma that tissue-resident MCs generally do not proliferate.

We found no significant difference in MC apoptosis between vehicle control- and CRH-treated NPs using quantitative tryptase/terminal deoxynucleotidyl transferase (TdT)-mediated dUTP nick end-labeling (TUNEL) double-immunofluorescence (Figure 3c). Thus, we can rule out that the changes in the number of hM-MCs merely reflect enhanced MC survival under the harsh conditions of serum-free organ culture. Therefore, we present evidence that CRH stimulates the intramucosal proliferation of native human tryptase+ M-MCs ex vivo and can expand the pool of tissue-resident MCs in human upper airway mucosa.

### 2.4. CRH-Induced hM-MC Effects in Human Upper Airway Mucosa Depended on Signaling Through CRH-R1

Importantly, both the increase in hM-MC number and their enhanced degranulation were mitigated by co-administration of antalarmin (Figure 2c,e). To exclude that this may have been due to potential off-target effects of antalarmin, we independently confirmed the result by CRH-R1 gene silencing in NPs ex vivo. Knocking down the receptor gene significantly counteracted the CRH-induced increase in both the total number of tryptase+ hM-MCs and percentage of degranulated hM-MCs in situ (Figure 3d,e). Therefore, ex vivo, CRH regulates native hM-MCs within NP tissue via CRH-R1-mediated signaling.

### 2.5. CRH Increased SCF Expression in NP Epithelium In Situ

SCF is a key growth factor for the maturation and proliferation of MCs [64]. We previously showed that CRH, cannabinoid receptor type 1 (CB1) antagonists, and CB1 siRNA each significantly increased SCF protein expression within the epithelium of human HFs ex vivo, which then stimulated the local differentiation of resident MC progenitors into mature tryptase+ skin MCs [11,53]. CB1 antagonists or CB1 siRNA have the same effect in NPs [50]. Therefore, we evaluated the effect of CRH stimulation on SCF protein expression in situ NPs using immunofluorescence.

As in human HF organ culture [53], CRH significantly increased SCF immunofluorescence intensity within the mucosal epithelium of organ-cultured NPs (Figure 4a), with double-immunofluorescence of SCF with AE1/3 (Figure 4b). However, some tryptase+ MCs also expressed SCF (Figure 4c), which is in line with previous reports that human mucosal-type MCs express SCF [65]. We noted a tendency for higher SCF expression within tryptase+ hM-MCs in NPs after CRH treatment compared with the vehicle control, but the difference was not significant (Appendix A).

### 2.6. SCF Was Required for CRH-Induced hM-MC Proliferation and Partially for MC Degranulation

The findings raised the question as to whether the CRH-induced effects on hM-MC proliferation and degranulation are SCF-dependent. The CRH-induced increase in the number of tryptase+ hM-MCs was mitigated by co-administration of anti-SCF (Figure 2c). Anti-SCF also partially inhibited CRH-induced hM-MC degranulation in NPs ex vivo (Figure 2e). The upregulation of SCF expression in NP epithelium by CRH was significantly mitigated by CRH-R1 gene silencing in NPs (Figure 4d).

### 2.7. CRH Sensitized hM-MCs to Further CRH Stimulation and Promoted a Pro-Inflammatory hM-MC Phenotype

As these findings would be of clinical relevance under, for example, conditions of stress-aggravated nasal allergy or chronic rhinosinusitis [4,66,67], we evaluated whether CRH upregulates its own receptor, sensitizing hM-MCs in a positive feedback loop to further CRH stimulation. Quantitative immunohistomorphometry of tryptase/CRH-R1 double-positive cells demonstrated that CRH significantly increased the expression of CRH-R1 on tryptase+ hM-MCs in situ (Figure 5a). A total of 15–50% of c-Kit+ MCs express CRH-R1 and around 15% of tryptase+ MCs express CRH-R1 in human NPs (Figure 5a,b). In contrast with tryptase+ MCs, CRH treatment did not affect the percentage of c-Kit/CRH-R1 double+ MCs (Figure 5b), suggesting that CRH primarily affects tryptase+ MCs in human nasal mucosa.

Tryptase is a pro-inflammatory trypsin-like protease stored within MC granules [55,68]. Tryptase promotes tissue inflammation via multiple pathways, including protease-activated receptors [68], activation of extracellular matrix-degrading enzymes, or recruitment of eosinophils/neutrophils [69]. Therefore, we also checked whether CRH increases tryptase expression within hM-MCs. As shown in Figure 5c, tryptase protein immunoreactivity was significantly upregulated by CRH. Taken together, the findings suggest that stimulation of MCs in the human nasal mucosa by the central neuroendocrine stress mediator CRH not only induces MC proliferation and degranulation, but also initiates a vicious circle whereby hM-MCs become sensitized to further CRH stimulation by enhanced CRH-R1 expression and attain an increasingly pro-inflammatory phenotype in situ.

### 2.8. Perceived Stress Increased MC Number and Degranulation in Murine Nasal Mucosa In Vivo via CRH-R1-Mediated Signaling

Finally, we examined whether perceived stress can increase the number and percentage of degranulated nasal mucosa MCs in an in vivo setting, and whether such an effect is CRH/CRH-R-dependent. To address this question, we conducted a limited pilot experiment using a well-established mouse model of restraint stress [51] and evaluated mM-MCs using quantitative toluidine blue histomorphometry. We also measured the serum cortisol and CRH levels on day 7 after 3 h of restraint stress in both stressed and sham-treated control mice. This confirmed the expected increase in serum cortisol and CRH levels in stressed mice (Appendix A) [51].

Both acute (once for 3 h) and chronic restraint stress (daily for 3 h over 7 days) significantly increased both the total number and the percentage of degranulated and histochemically detectable mM-MCs in the stressed mice compared with sham-stressed control animals (Figure 6a–c). Importantly, these effects of perceived stress on mM-MCs in vivo were significantly reduced by nasal application of antalarmin for 1 day or for 7 consecutive days (Figure 6b,c).

Although CRH-R1 has other ligands besides CRH, namely, urocortin, another important neuroendocrine stress mediator [8,70], these preliminary results strongly suggested that our CRH stimulation data generated in organ-cultured human NPs translate to the in vivo response of MCs in the nasal mucosa, and that the CRH-R1-mediated perceived stress responses of M-MCs may be reasonably conserved between the two species.

## 3. Discussion

With this study, we advance the field by demonstrating the following: (1) hM-MCs predominantly expressed CRH-R1 in situ, (2) CRH increased the number of tryptase+ hM-MCs by stimulating proliferation rather than inducing hM-MC maturation, (3) CRH induced hM-MC degranulation, (4) all of these effects were CRH-R1-dependent, (5) CRH also upregulated SCF expression within the human upper airway epithelium in a CRH-R1-dependent manner, (6) CRH-induced hM-MC degranulation and proliferation were SCF-dependent, (7) CRH sensitized hM-MCs to further CRH stimulation and promoted development of an increasingly tryptase+ pro-inflammatory hM-MC phenotype in situ, and (8) perceived stress increased the number and degranulation of MCs within the nasal mucosa of mice in vivo and this effect was mitigated by CRH-R1 blockade. Given the recognized central role of hM-MCs in nasal allergy [46,47,71] and the aggravation of nasal allergy by psychoemotional stress [4,66], the current results are clinically important because they identify one plausible and pharmacologically targetable neuroendocrine mechanism. Our data show that agents that antagonize CRH-R1-mediated signaling are promising tools for clinical intervention in stress-aggravated, MC-dependent mucosal inflammation, namely, nasal allergy.

Our data strongly argue in favor of the concept that CRH acts primarily as a potent human MC secretagogue, not only in human skin [11] but also in the upper airway mucosa. This is in line with the sentinel function of MCs [13,17] and their involvement in the pathobiology of atopic dermatitis, psoriasis, and alopecia areata, which are skin diseases that can all be exacerbated by stress [19,42,72,73]. Moreover, our finding that the epithelial production of SCF, arguably the most important MC growth factor [64,74], in human nasal mucosa is under neuroendocrine control by a key stress-mediating neurohormone (i.e., CRH) echoes our previous findings in human skin [11]. This further encourages more rigorous exploration of how locally produced neurohormones are recruited to control the expression and secretion of evolutionarily much younger growth factors in human tissues, namely, under conditions of perceived stress [9].

We also confirm that, in contrast to conventional wisdom that tissue-resident MCs typically do not proliferate, hM-MCs can proliferate within (inflamed) nasal mucosa [75], just as CT-MCs do in inflamed human skin in vivo [63]. That this MC proliferation in situ is further stimulated by CRH makes the development of pharmacological intervention strategies to curb a clinically undesired expansion of the intramucosal MC pool under conditions of psychoemotional stress even more compelling.

Interestingly, CRH also increased the diameter of tryptase+ granules in hM-MCs. If the process of MC degranulation is preceded by a fusion of secretory granules, and increased secretory granule size indicates that MCs are preparing for degranulation [61,76], this phenomenon further corroborates that CRH is a potent secretagogue for primary, unmanipulated hM-MCs within their natural tissue habitat. We also showed that CRH enhances tryptase expression within individual hM-MCs, mirroring a phenomenon we previously observed in perifollicular human CT-MCs (hCT-MCs) alopecia areata skin lesions [63]. As tryptase is a pro-inflammatory protease [55,68], psychoemotional stress may aggravate MC-dependent mucosal inflammation via CRH-induced tryptase secretion.

In contrast to the effect of CRH on hCT-MCs [11], we did not find convincing evidence that CRH stimulates the maturation of MCs from resident progenitor cells in human nasal mucosa ex vivo. This novel, differential response of hM-MCs and hCT-MCs to the same neuroendocrine stimulus points to additional functional differences between hCT-MCs and hM-MCs beyond their recognized distinct protease expression profiles [55,77,78].

The CRH-R1 antagonist antalarmin and anti-SCF both abrogated CRH-induced hM-MC proliferation and degranulation in inflamed human nasal mucosa, NPs. This raises the question as to whether these effects also extend to mucosa MC populations in other organs and whether CRH-R antagonists and/or anti-SCF can be exploited in other MC-dependent inflammatory and allergic diseases aggravated by psychological stress [79], ranging from allergic rhinoconjunctivitis, allergic asthma and food allergy to inflammatory bowel disease, anaphylaxis, and urticaria [47,71,80,81].

Although the preliminary pilot data from our in vivo mouse experiment require confirmation and translation to the human system, they encourage the full exploration of nasally applied CRH-R1 antagonists in the future therapy of stress-induced MC-dependent nasal mucosa inflammation, in line with previous reports on the therapeutic benefits of CRH-R1 antagonists in inflammatory bowel disease or bladder hyperactivity [52,82].

## 4. Materials and Methods

### 4.1. Reagents

CRH was purchased from Sigma-Aldrich (St. Louis, MO, USA), SCF-neutralizing antibody (#AB-255-NA) from R&D Systems (Minneapolis, MN, USA), and antalarmin from Cayman Chemical (Ann Arbor, MI, USA).

### 4.2. Human Nasal Polyp Organ Culture

Human NP samples were obtained from patients with chronic sinusitis (15 patients, 8 men and 7 women), aged 26–80 years (average age = 54.4 years; Appendix A) who were undergoing endoscopic sinus surgery for nasal obstruction. Human tissue collection and handling was performed according to the Declaration of Helsinki guidelines with approval from the institutional research ethics committee (Osaka City University Independent Ethics Committee, ethical permit numbers 2511, approved on 12 March 2013) and written informed consent from the patients. NP organ culture was performed as described previously [50]. Briefly, freshly isolated NP tissue (soaked in saline solution to prevent drying out) was cut into small pieces (6 mm diameter) within 1 h of surgery using a 6 mm biopsy punch (Kai Industries Co., Ltd., Gifu, Japan) and maintained in supplemented serum-free William’s E medium [11,50,53,83]. NPs were pre-incubated overnight to allow them to adapt to culture conditions. Afterward, the NPs were treated with vehicle (supplemented Williams’s E medium), 10^−7^ M CRH [50], or 10^−7^ M CRH, and/or anti-SCF (1 μg/mL) [50], 10^−7^ M CRH, and/or antalarmin (10^−6^ M), CRH-R1-specific antagonists [84]. These compounds were co-administered with CRH and cultured for 5 days. Medium exchange was performed every other day. After NP organ culture for the indicated duration, the tissue was processed for paraffin sectioning, immunohistochemistry/immunofluorescence, electron microscopy, or CRH-R1 gene knockdown to clarify the effect of CRH on the mucosal-type MC-rich human NP sample. Each experiment was independently repeated with NP samples from 4–7 different patients.

### 4.3. Toluidine Blue Histochemistry

Toluidine blue staining was applied to investigate the MC localization and morphology. All deparaffinized sections were stained with 1% toluidine blue (pH 8.9; Merck, Darmstadt, Germany) for 3 min at room temperature (RT) [50].

### 4.4. Immunohistochemistry/Immunofluorescence Microscopy

For detection of c-Kit, tryptase, chymase, SCF, CRH, CRH-R1, and CRH-R2 antigens (Table 1), paraffin-embedded sections were immunoassayed following established protocols [11,50,63,85]. For the detection of c-Kit, tryptase, chymase, and CRH, all deparaffinized sections underwent antigen retrieval using an autoclave at 121 °C for 15 min and 103.7 kPa. After pre-incubation with 0.5% Triton-X in phosphate-buffered saline (PBS) containing 6% bovine serum albumin at RT for 30 min, sections were incubated overnight at 4 °C with primary antibodies diluted in PBS. The following primary antibodies were used: mouse anti-human tryptase (Abcam, Cambridge, U.K.) at 1:500, rabbit anti-human c-Kit (CD117; Cell Marque, CA, USA) at 1:200, goat anti-human chymase (Proteintech, Chicago, IL, USA) at 1:500, and rabbit anti-human CRH (Proteintech) at 1:50. Next, the sections were incubated with goat biotinylated antibodies against rabbit or mouse IgG (Jackson ImmunoResearch Laboratories, West Grove, PA, USA) at 1:200 for 45 min at room temperature (RT). The sections were then treated with the alkaline phosphatase-based avidin-biotin complex (Vectastain Elite ABC kit; Vector Laboratories, Burlingame, CA, USA) and the expression of the antigens visualized with Fast Red (Sigma-Aldrich, St. Louis, MI, USA).

Immunofluorescence was performed to detect SCF, CRH-R1, and CRH-R2. All deparaffinized sections underwent antigen retrieval using an autoclave at 121 °C for 15 min (with tris-ethylenediaminetetraacetic acid or sodium citrate buffer) or treated with proteinase K working solution for 15 min at 37 °C. After antigen retrieval, the sections were incubated overnight at 4 °C with primary antibodies diluted in PBS. The following primary antibodies were used: rabbit anti-human SCF (Abcam) at 1:50, goat anti-human CRH-R1 antibody (Abcam) at 1:200, and rabbit anti-human CRH-R2 (Life Technologies, Carlsbad, CA, USA) at 1:200. This was followed by incubation with goat anti-rabbit or rabbit anti-goat IgG Alexa Fluor 488 (Thermo Fisher Scientific, Waltham, MA, USA) diluted 1:200 in PBS for 45 min at RT. To perform double-immunofluorescence labeling, we used the appropriate primary antibody and secondary antibodies conjugated to the correct fluorophore (Appendix A). For tryptase and TUNEL double immunofluorescence, after immunostaining for tryptase, the sections were immersed in 1× TdT labeling buffer from the TACS 2 TdT-Flour In Situ Apoptosis Detection Kit (Trevigen, Gaithersburg, MD, USA) for 5 min. Next, the samples were incubated with Labeling Reaction Mix in the presence of terminal deoxynucleotidyl transferase for 60 min at 37 °C. After incubation with stop buffer for 5 min at RT and an additional wash with PBS, TUNEL-positive cells were visualized by step-fluorescein for 20 min at RT. Nuclear labeling was performed using 4′,6-diamidino-2-phenylindole (DAPI; Thermo Fisher Scientific). For negative controls, the appropriate primary antibodies were omitted from the procedure. The specificity of CRH, CRH-R1, and CRH-R2 immunostaining was measured on intact human skin sections and mouse brain sections (positive control), which clearly demonstrated positive immunoreactivity in the expected areas. For SCF and AE1/3 immunostaining, an intact human skin sample was used as a positive control. When counting MCs, MCs were classified as degranulated when ≥5 extracellularly located metachromatic granules could be detected [50] histochemically at high magnification (400×) by light microscopy (visual field). The number of degranulated and total human M-MCs within the lamina propria of a human NP per visual field was counted, and at least 15 visual fields per NP were evaluated.

### 4.5. Quantitative (immuno)histomorphometry

The immunoreactivity of CRH, CRH-R1, CRH-R2, tryptase, and SCF was quantified by assessing the immunoreactivity in defined reference areas and by quantitative immunohistomorphometry using ImageJ software (National Institutes of Health, Bethesda, MD, USA).

### 4.6. CRH-R1 Gene Knockdown

*CRH-R1* gene silencing in organ-cultured human NPs was achieved using a previously reported method [50]. All reagents required for transfection (human CRFRI siRNA (sc-39914), control (scrambled, SCR) siRNA (sc-37007), siRNA transfection reagent (sc-29528), and siRNA transfection medium (sc-36868)) were obtained from Santa Cruz.

Human NP transfection was performed according to the manufacturer’s protocol. Briefly, freshly isolated human NPs were kept in cold William’s E medium immediately before transfection. During transfection, CRH-R1-specific siRNA or control siRNA (6 μL) and siRNA transfection reagent (6 μL) were mixed in transfection medium (500 μL) in each well (24-well plate). After careful washing, human NPs were applied to each well in an incubator for 8 h, after which the medium was replaced with supplemented William’s E medium. Human NPs were fixed in 4% paraformaldehyde/PBS 24 h following transfection. Effective mucosa epithelial CRH-R1 knockdown was demonstrated by a significant downregulation of CRH-R1 immunoreactivity by 56% in the nasal mucosa epithelium (Appendix A).

### 4.7. Restraint Stress Mouse Model

We used a previously described mouse model of restraint stress [51]. We worked with 12-week-old female C57BL/6 mice purchased from Japan SLC (Shizuoka, Japan). They were allowed to acclimatize for 7 days in our animal facility. All mice were single-housed with a 12/12 h light/dark cycle, and food and water were available on ad libitum. All protocols were approved by the Institutional Animal Care and Use Committee at Osaka City University. Mice were randomly assigned to experimental groups and restrained in a DecapiCone device (Braintree Scientific, Inc., Braintree, MA, USA) with all limbs positioned flat underneath the body to minimize physical pain. The end of the plastic tube was rolled shut and secured with clips. After 3 h of restraint, mice were placed back into their home cage. Restraint stress was applied once daily for 7 consecutive days. On days 1 and 7, immediately after restraint stress, all mice were sedated, and nasal mucosa were collected. We evaluated the M-MCs after both acute stress (1 day of restraint only) and chronic stress (7 consecutive days of restraint stress). In vivo studies were conducted in the following 4 treatment groups for both the acute and chronic stress groups: (1) without restraint stress (control group; *n* = 4), (2) exposed to restraint stress (stress group; *n* = 4), (3) exposed to restraint stress treated with nasal application of CRH-R1 antagonist antalarmin (5 μg/g/day) for 7 days during the stress exposure (stress + antalarmin group; *n* = 5), and (4) without stress treated with nasal application of antalarmin for 7 days (antalarmin group; *n* = 4). All treatments were given 30 min prior to restraint.

For antalarmin, the reported concentration of 20 µg/g of body weight (diluted by polyethylene glycol) was used as a reference [86]. As we performed nasal application in this study, we administered 5 µg/g (30 µL) antalarmin to each mouse each day for 7 days. Control mice had a nasal application of the vehicle (polyethylene glycol). The number of degranulated and total MCs within the lamina propria of the mouse nasal mucosa per visual field was counted using toluidine blue histochemistry, and at least 15 visual fields per mouse were evaluated. This experiment was repeated twice.

### 4.8. Measurement of Plasma Corticosterone and CRH Levels

Mice were decapitated rapidly at the end of the stress period, and blood (about 1 mL) was collected and centrifuged (12,000 rpm, 10 min, and 4 °C) to separate serum. The serum samples were stored at −80 °C until hormone assay.

Cortisol levels in serum were determined using the cortisol (human/mouse/rat) ELISA Kit (BioVision, Milpitas, CA, USA) and CRH levels in serum were determined using the YK131 Mouse/Rat CRF-HS ELISA Kit (Yanaihara Institute Inc, Sizuoka, Japan) according to the manufacturer’s instructions.

### 4.9. Statistical Analysis

Data were analyzed using either the Mann–Whitney *U* test for unpaired samples or one-way analysis of variance (ANOVA) with the Bonferroni multiple comparison test in Prism 8.0 software (GraphPad Prism; GraphPad Software, San Diego, CA, USA). Data are presented as mean ± standard error of the mean (SEM).

## 5. Conclusions

Our study introduces CRH and SCF as key players in the aggravation of MC-dependent neuroinflammation in human nasal mucosa by perceived stress, and the findings suggest that intranasal CRH-R1 antagonists can counteract this phenomenon in vivo.

## Figures and Tables

**Figure 1 ijms-22-02773-f001:**
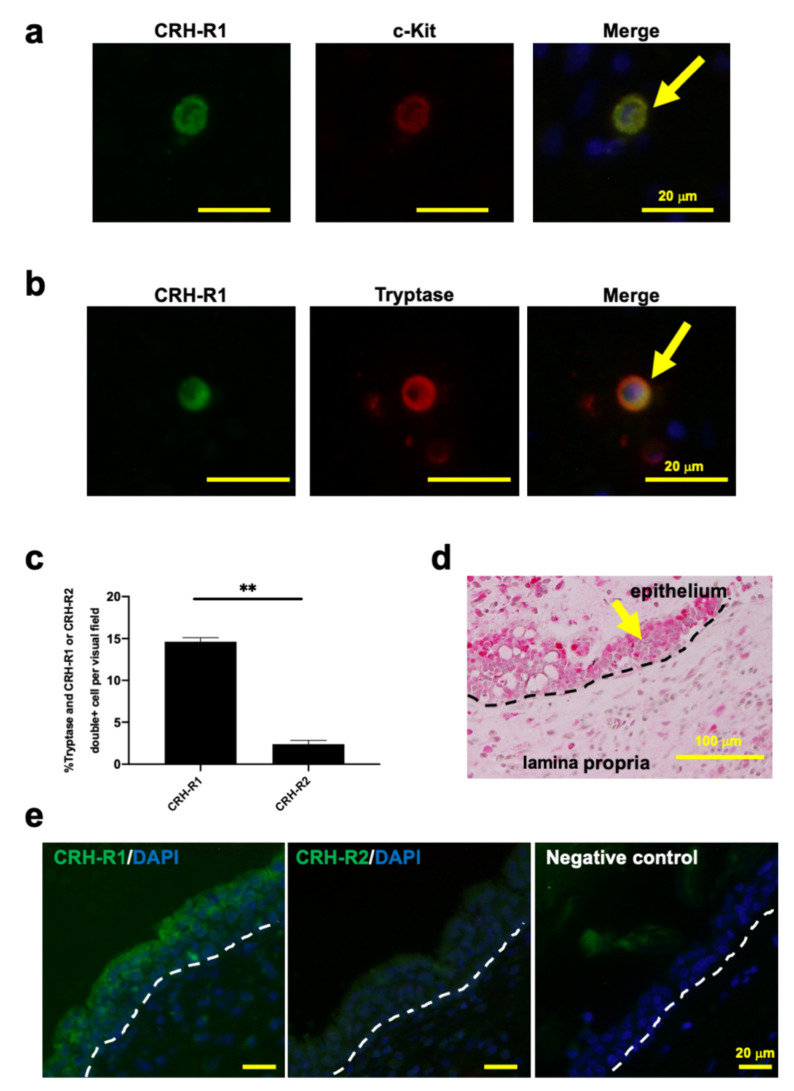
Human nasal mucosa mast cells (hM-MCs) and the mucosal epithelium in nasal polyps (NPs) express corticotropin-releasing hormone receptor type 1 (CRH-R1). (**a**) CRH-R1 expression on c-Kit+ or (**b**) tryptase+ hM-MCs in situ. An arrow denotes double+ MCs. (**c**) Percentage of tryptase and CRH-R1 or corticotropin-releasing hormone receptor type 2 (CRH-R2) double+ MCs. *n* = 5. (**d**) CRH expression within the mucosal epithelium in NPs (an arrow). (**e**) CRH-R1 and CRH-R2 expression within the mucosal epithelium in NPs. Scale bar = 20 µm. Error bars indicate the standard error of the mean (SEM). ** *p* < 0.01. MC, mast cell; hM-MCs, human nasal mucosa MCs; NPs, nasal polyps; CRH, corticotropin-releasing hormone; CRH-R1, CRH receptor type 1; CRH-R2, CRH receptor type 2; DAPI, diamidino-2-phenylindole.

**Figure 2 ijms-22-02773-f002:**
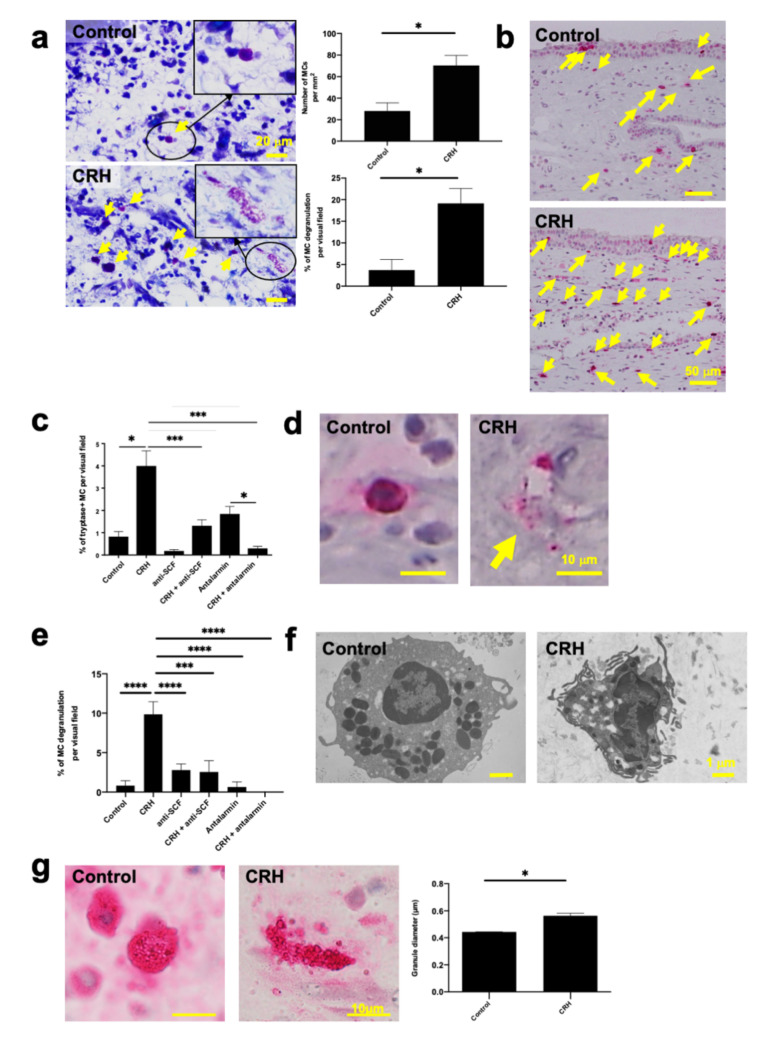
CRH increased tryptase+ hM-MC numbers and stimulated their degranulation. (**a**) Toluidine blue histochemistry with 1 day organ-cultured human NPs treated with vehicle or CRH (10^−7^ M). An arrow denotes hM-MCs in the lamina propria. Top, non-degranulated MCs in vehicle-treated NPs. Bottom, degranulated MCs in CRH-treated NPs. *n* = 5; scale bar = 20 µm. (**b**) Tryptase immunohistochemistry with 1 day organ-cultured human NPs treated with vehicle or CRH (10^−7^ M). Scale bar = 50 µm. (**c**) Quantitative immunohistomorphometry of tryptase+ cells in the lamina propria of organ-cultured NPs with CRH, anti-SCF (stem cell factor neutralizing antibody; 1 μg/mL), and/or antalarmin (10^−6^ M); *n* = 6. (**d**) Tryptase immunohistochemistry. An arrow denotes MC degranulation induced by CRH. Scale bar = 10 µm. (**e**) Percentage of degranulated MCs treated with CRH, anti-SCF (1 μg/mL), and/or antalarmin (10^−6^ M) analyzed by tryptase immunohistochemistry; *n* = 6. (**f**) Electron microscope images of CRH-induced MC degranulation. Scale bar = 1 µm. (**g**) Diameter of tryptase+ hM-MC granules; *n* = 6; scale bar = 10 µm. Error bars indicate SEM. * *p* < 0.05, *** *p* < 0.001, **** *p* < 0.0001. MC, mast cell; hM-MCs, human nasal mucosa MCs; NPs, nasal polyps; CRH, corticotropin-releasing hormone; anti-SCF, stem cell factor neutralizing antibody.

**Figure 3 ijms-22-02773-f003:**
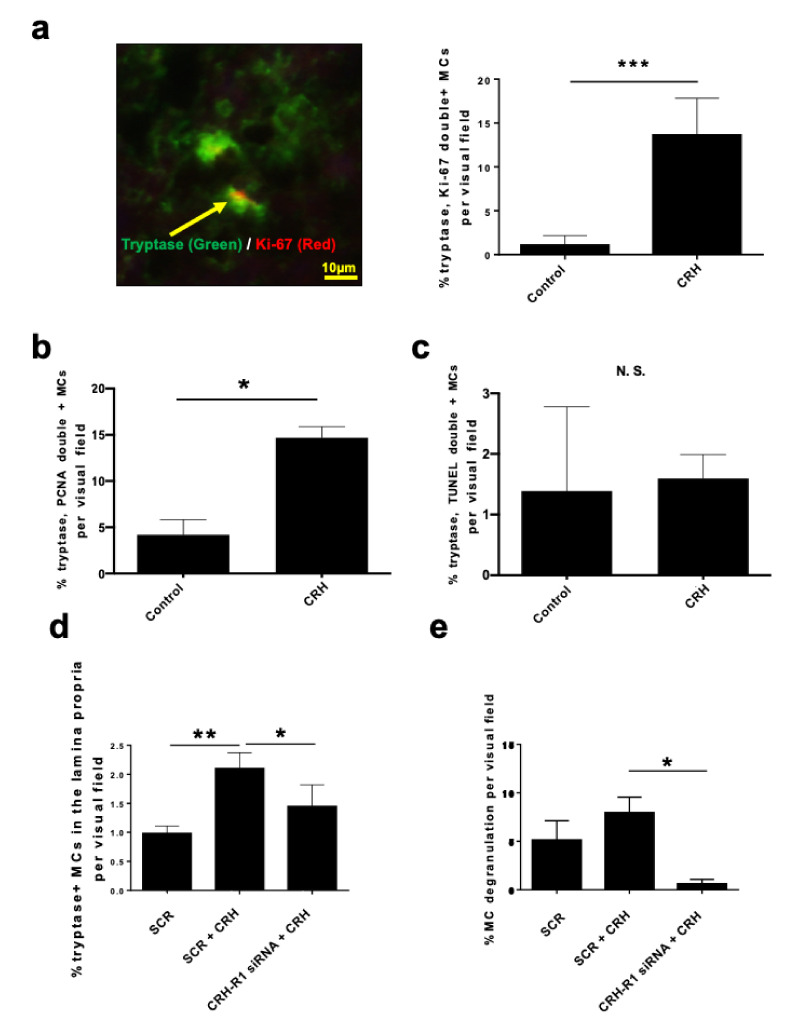
CRH increased tryptase+ hM-MC proliferation but not the apoptosis of hM-MCs. CRH-R1 gene silencing showed that CRH-R1 small interfering RNA (siRNA) inhibited the increase in tryptase+ hM-MC number and degranulation by CRH in situ. (**a**) Tryptase/Ki-67 double immunofluorescence. The arrow denotes tryptase and Ki-67 double+ cells within human nasal polyps treated with CRH; *n* = 6; scale bar = 10 µm. (**b**) Quantitative immunohistomorphometry of tryptase/proliferating cell nuclear antigen (PCNA) double+ cells; *n* = 4. (**c**) Quantitative immunohistomorphometry of tryptase/terminal deoxynucleotidyl transferase-mediated dUTP nick end-labeling (TUNEL) double+ cells; *n* = 4. (**d**) Quantitative immunohistomorphometry of tryptase+ cells in the lamina propria of organ-cultured NPs with CRH-R1 siRNA-treated human NPs; *n* = 4. (**e**) Percentage of degranulated hM-MCs in the lamina propria with CRH-R1 siRNA-treated human NPs; *n* = 4. Error bars indicate SEM. * *p* < 0.05, ** *p* < 0.01, *** *p* < 0.001, N.S. = not significant. MC, mast cell; hM-MCs, human nasal mucosa MCs; NPs, nasal polyps; CRH, corticotropin-releasing hormone; CRH-R1, CRH receptor type 1; CRH-R1 siRNA, CRH-R1 siRNA-treated NPs; SCR, scrambled siRNA-treated NPs; PCNA, proliferating cell nuclear antigen; TUNEL, terminal deoxynucleotidyl transferase-mediated dUTP nick end-labeling.

**Figure 4 ijms-22-02773-f004:**
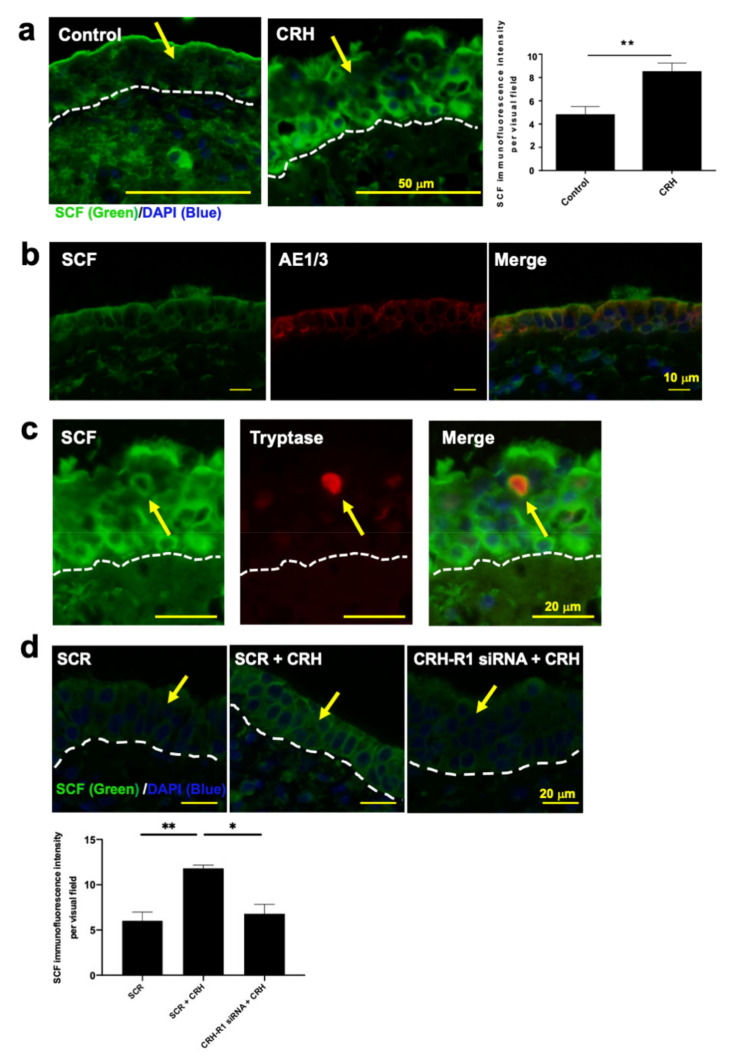
CRH increased stem cell factor (SCF) expression in the mucosal epithelium in NPs. (**a**) SCF immunofluorescence of organ-cultured human NPs treated with vehicle or CRH (10^−7^ M). CRH significantly increased SCF expression within the epithelium of 1 day organ-cultured human NPs. The arrow denotes SCF-positive immunoreactivity within the epithelium. *n* = 6; scale bar = 50 µm. (**b**) Double immunofluorescence for AE1/3/SCF and (**c**) tryptase/SCF of CRH-treated NPs. Most of the SCF+ cells were epithelial cells. Some tryptase+ hM-MCs within the epithelium expressed SCF. The arrow denotes SCF+ hM-MCs; *n* = 4 in (**b**) and 6 in (**c**). Scale bar = 10 µm in (**b**), 20 µm in (**c**). (**d**) SCF immunoreactivity in NP epithelium with CRH-R1 siRNA-treated human NPs. The arrow denotes SCF-positive immunoreactivity within the epithelium. *n* = 4; scale bar = 20 µm. Error bars indicate SEM. **p* < 0.05, ** *p* < 0.01. MC, mast cell; SCF, stem cell factor; hM-MCs, human nasal mucosa MCs; NPs, nasal polyps; CRH, corticotropin-releasing hormone; CRH-R1, CRH receptor type 1; CRH-R1 siRNA, CRH-R1 siRNA-treated NPs; SCR, scrambled siRNA-treated NPs; DAPI, diamidino-2-phenylindole.

**Figure 5 ijms-22-02773-f005:**
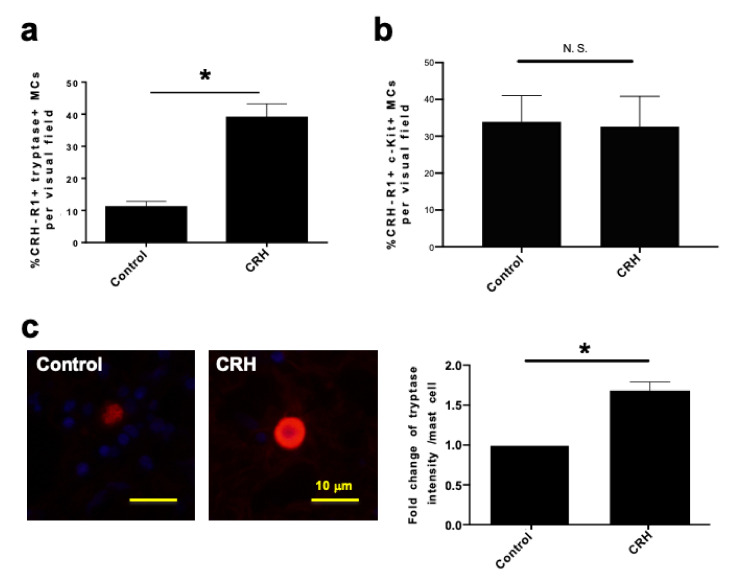
CRH increased the expression of CRH-R1 on tryptase+ hM-MCs and upregulated tryptase immunoreactivity within the hM-MCs. Percentage of (**a**) tryptase+ and (**b**) c-Kit+ MCs with CRH-R1 in the lamina propria of organ-cultured human NPs, analyzed by double immunofluorescence of tryptase or c-Kit/CRH-R1. *n* = 7 in (**a**) and 4 in (**b**). (**c**) Tryptase immunoreactivity within the hM-MCs in NPs treated with vehicle or CRH. CRH treatment significantly increased tryptase immunoreactivity within hM-MCs. *n* = 5; scale bar = 10 µm. Error bars indicate SEM. * *p* < 0.05, N.S. = not significant. MC, mast cell; hM-MCs, human nasal mucosa MCs; NPs, nasal polyps; CRH, corticotropin-releasing hormone; CRH-R1, CRH receptor type 1.

**Figure 6 ijms-22-02773-f006:**
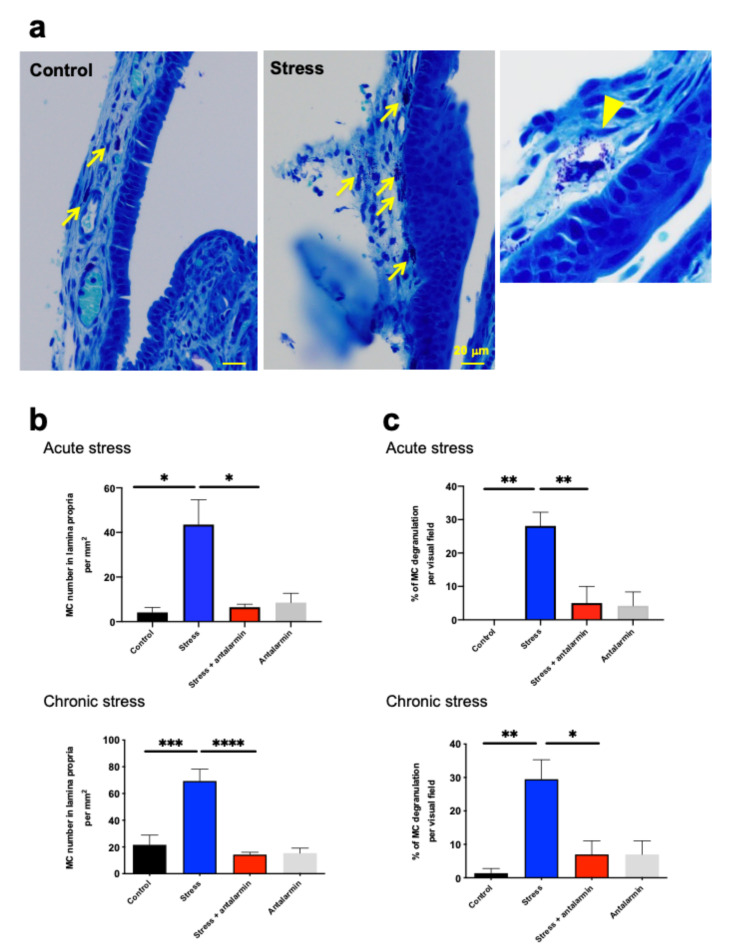
Restraint stress increased mice nasal mucosa MC (mM-MC) numbers and degranulation. (**a**) Toluidine blue histochemistry of murine nasal mucosa. The arrow denotes mM-MCs in the lamina propria, and the arrowhead denotes a degranulated mM-MC. Scale bar = 20 µm. (**b**) Quantitative immunohistomorphometry of mM-MC. (**c**) The percentage of degranulated mM-MCs in the nasal mucosa after restraint stress and nasal application of antalarmin. Both acute and chronic restraint stress significantly increased mM-MC numbers in the nasal mucosa and stimulated their degranulation. The increased number of mM-MCs and their degranulation in stressed mouse nasal mucosa was significantly inhibited by nasal application of antalarmin. Control group, *n* = 4 (without stress); stress group, *n* = 4; stress + antalarmin (5 μg/g/day) group, *n* = 5; and antalarmin group, *n* = 4 (without stress). Error bars indicate SEM. * *p* < 0.05, ** *p* < 0.01, *** *p* < 0.001, **** *p* < 0.0001. MC, mast cell; mM-MCs, mice nasal mucosa MCs.

**Table 1 ijms-22-02773-t001:** List of primary antibodies.

Antigen	Isotype	Dilution	Supplier	Cat.#	RRID
CRH	rabbit	1:50	Prointech	10944	AB_2084279
CRH-R1	goat	1:200	Abcam	ab77686	AB_1566096
CRH-R2	rabbit	1:200	Invitrogen	720291	AB_2633243
Tryptase	mouse	1:500	Abcam	ab2378	AB_303023
C-Kit	rabbit	1:200	Cell Marque	117R-16	AB_1159085
Chymase	goat	1:500	Abcam	ab111239	AB_10863662
PCNA	rabbit	1:200	Abcam	ab92552	AB_10561973
Ki-67	rabbit	1:20	Abcam	ab16667	AB_302459
SCF	rabbit	1:100	Abcam	ab64677	AB_1861346
AE1/3	goat	1:400	Abcam	ab86734	AB_10674321

CRH, corticotropin-releasing hormone; CRH-R1, CRH receptor type 1; CRH-R2, CRH receptor type 2; PCNA, proliferating cell nuclear antigen; SCF, stem cell factor.

## Data Availability

All data are available on request from the authors.

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
