# Peer review of "Stress and Nasal Allergy: Corticotropin-Releasing Hormone Stimulates Mast Cell Degranulation and Proliferation in Human Nasal Mucosa"

_ijms, 2021, doi:10.3390/ijms22052773_

Round 1
Reviewer 1 Report
Paper entitled:” Stress” and nasal allergy: Corticotropin-releasing hormone stimulates mast cell degranulation and proliferation in humannasal mucosa” (ijms-1130923) presents an interesting and globally important problem regarding the understanding of the mechanism generation of stress-induced nasal allergies. Obtained results seem very interesting and novel. However, there are some issues which should be addressed before publication in IJMS.
Figure 1d: please mark the presence of CRH on the photos.
Figure 4a: please correct the description on the y-axis of the graph
Materials and methodology
- please insert catalog numbers and RRID for all antibodies used
4.4 L431:
I cannot imagine autoclaving tissues at a constant temperature of 121 degrees for 15 minutes. Autoclaving ensures constant temperature and pressure conditions.
Please write what the tissues were immersed in to protect them from drying out?
English should be improved by native English speaker
Reviewer 2 Report
This is a great study proving stress-induced mast cell-dependent nasal mucosa inflammation.
However, if you show changes in symptoms such as sneezing and/or rubbing count in mice after restraint stress or administration of antalarmin, your study may be more reasonable.
In addition, why did you use organ culture with nasal polyp tissue, not nasal mucosa from patients with allergic rhinitis? Did all patients with chronic sinusitis with nasal polyps have allergy or eosinophilic inflammation? It should be concerned that the pathophysiology of CRS may not exhibit mast cell-dependent allergic reaction or type 2 immune response,
